# VUOIŊŊALAŠVUOHTA—Sámi Spirituality, Yoik and Its Relations †

**Tuula Sharma Vassvik**

Independent Researcher, 0482 Oslo, Norway; sharmavassvik@gmail.com; Tel.: +47-930-30877
† North Sámi for spirituality.

**Abstract:** The subject of identity is important in today's political landscape. This article explores the way in which indigenous identity in particular is a contested subject, taking into account the way indigeneity in itself was, and still is, created within colonial contexts. The "validity" of indigenous peoples and their political aims, as well as their right to live according to their own cultural paths, will often be determined according to racist ideas connected to authenticity and its stereotypical demands. Such concepts can furthermore turn inward, disconnecting indigenous peoples from their own heritage. How ideas of authenticity affect indigenous individuals and their processes of identification serves as a central question within this text. Central to the Standing Rock movement was the focus on spirituality and religion grounded in Lakota traditions and other indigenous cultures. The text accounts for how these practices affected Marielle Beaska Gaup, Sámi artist, activist, and mother, especially through her experiences as a *juoigi*, a traditional Sámi yoiker. The ever-present singing and drumming at camp, chiefly during the summer, tied the mundane and ritualistic together, a characteristic mirroring traditional Lakota and Sámi ways of life, in which the spiritual at times seem to be an integral part of daily life. Building upon Marielle's observations, the text looks at the way indigenous people's relationships with spiritual traditions can affect processes of identification, and how indigenous identity can be intimately link to its spiritual heritage. This article employs indigenous methodologies, centering research on Sámi and indigenous perspectives, values and agendas. Marielle's reflections contribute to the exploration of the connections between spirituality and Sámi identities; furthermore, they enable us to connect ideas about moving beyond the authoritarian ideals of "authentic identities", through re-centering on indigenous experiences and processes of identification My main source is Marielle's interview and articles based on interviews with people from Standing Rock The analysis centers on Marielle's thoughts together with my own, with support from indigenous researchers bringing their own knowledge about identity and spirituality forward.

**Keywords:** identity; spirituality; yoik; indigenizing; decolonizing; recreating; music; religion; authenticity; indigenous methodologies; Standing Rock; activism; traditional knowledge; relationality; reconnecting

## 1. Introduction

Marielle Gaup Beaska came to Standing Rock to stand with the Lakota Nation of Turtle Island. Standing Rock, the name of the local reservation, but also the name of the water protector camp, was located by Lake Oahe (the Missoury River), North Dakota, from the springtime in 2016 until springtime in 2017. The camp was set up to protect the local drinking water from a pipeline and with time came to symbolize Native American Resistance, environmentalism and the fight against eco-racism worldwide. It became well known internationally for its many participants, the indigenous people and allies who came from all around the world, and its focus on peaceful ways of protesting.

Marielle is a Sámi woman from Guovdageaidnu (Kautokeino), a *juoigi* (a yoiker, yoik is a Sámi traditional form of singing), an activist and a mother. Here, I will use the word yoik (*luothi*) and the

words yoiking about the verb (*juoigan*). The history of yoik is a complex one. Six-thousand-year-old rock carvings and written sources from the 12th, 13th, 17th and 18th century support the fact that yoik has been a part of the Sámi societies for a very long time (Graff 2014, p. 67). Many sources talk about the ritual practice of yoiking, but also the everyday importance of it. How kids often learned to yoik before they learned to speak, for example (Graff 2004). Yoik has also been an important way of communicating and connection for Sámi people in the context of Norwegianisation. However, the strict laws forbidding anyone to yoik from the 17th century and the persecution of Sámi *noaiddit* (Sámi who had contact with other realms, could influence beings and help when someone were unwell or had a problem) have had a great effect on Sámi people. These views on yoik spread to the Christianized Sámi populations and strong beliefs about yoik being a devilish practice have made many Sámi wary of yoik, especially the older generations, and in Sámi core areas. The relationship to yoik is constantly changing, however, as will become apparent in the sections below.

Marielle is the main interviewee in this text. The material builds on my master's thesis "Standing Rock as a Place of Learning—Strengthening indigenous Identities", where Marielle also plays a big role. For my thesis, I interviewed several other activists from Standing Rock, Zintkala Mahpia Win Blackowl is one of them, and also a part of this text. I have also cited other interviews with activists from Standing Rock and spoken with and cited other *juoigit* (yoikers), including Sten Jörgen Stenberg and my father, Torgeir Vassvik. Conversations with friends such as Kristin Solberg, Andreas Daugstad Leonardsen, my mother Neena Sharma Uhre, other relatives and friends along the way have been insightful and helpful. Through these accounts and conversations, I have tried to answer some of my own questions about Sámi spirituality, *vuoiŋŋalashvuohta*, and identity, and ask some new ones.

Here, I will go into relations between yoik and spirituality, identity being central to both of these themes together with a focus on connectivity. Identification is, in my experience, a process and a way of connection (Somerville 2011). In a world where a disconnection seems to be the norm, indigenous people working towards a closer relationship to land, family, community and spirituality is sometimes an effort that is misunderstood by members of the majority culture. Here, I will talk about the way processes of identification can lead to a sense of belonging, wellbeing and a stronger sense of connection. In what way is yoik, an important pillar of Sámi culture, instrumental in this ongoing process for Marielle and other yoikers? How has the role of yoik changed within Sámi communities?

Marielle and I met for the first time in Minneapolis, December 2017, preparing to go to Standing Rock. We were part of the same group of Sámi activists going there to help out in the ways we could and show our solidarity with the water protectors there. The movement gained a surprising amount of media-attention compared to many indigenous and/or environmentalist movements today and in the past. Here is an account from Cannupa Hanska Lugar, an artist, born on the Standing Rock reservation. His reflections put into words some of my thoughts and show how important this movement was for many Native Americans.

> Everybody came in hoping to experience something new, something profound. But when they got there, they realized they're not a part of something new, they've just been absorbed into something that is much older than the entire country. That's incredibly humbling.
>
> ( . . . ) The big difference is that I think [people have] had the opportunity to encounter us not as a mystic, romantic other. It's just like, "Dude, we're just human beings." What does "Lakota" mean in English? It literally means "the people." ( . . . ) This is why we say this is not a protest, why we are water protectors. We're not just in protest of a pipeline. What we are trying to do is maintain a cultural practice. This is our culture. It's a part of our society.
>
> ( . . . ) The amazing thing is that whether you were Native or not, what we witnessed up there is the awakening of a giant that has been sleeping. It's the power of us as living things—rather than us waiting for somebody to save us. It was so grass roots. Native people have never been subject to that amount of solidarity. It left everybody awestruck. And the number of Native people coming together, nothing like this has been seen since the 19th

century. Tribes that had previously been enemies, coming together—there's no way for me to describe to you what that means. It's far too profound[1].

There was something unique about Standing Rock that I think people could pick up on just by following the movement online. Many were attracted by the survivance that shone through via videos, music, articles and posts from Standing Rock that were spread online. Survivance is a term firstly used in the context of Native American studies by the Anishinaabe[2] cultural theorist Gerald Vizenor. It describes a life of active survival, of thriving as indigenous peoples, strengthening and recreating one's culture and identity freely without the confinements of stereotypes and oppressive authenticity (Sissons 2005). Oceti Sacowin at Standing Rock in many ways became a "liminal space" where people felt free to live lives of active survivance.

Hanska Lugar touches on several important points about the camp as a continuation of old traditions, worldviews, and with that the struggle for sovereignty and the right to a nondestructive and close relationship to the land and the water for everyone, stood out as an eloquent way of putting some of my own impressions into words. Personal and communal identity stand as grounding pillars in the work of indigenous activists, community workers, politics, artists and researchers all over the world. Here, I will talk about identity in a relational sense. With support from indigenous methodology, I delve into some of the ways that identity can be seen as connected to land, tradition, spirits and people.

*1.1. Colonialism in Sápmi and Decolonizing and Indigenizing Responses*

A: Mr. Cox, Spencer. For the last hundred and eighteen years, I have lived in your world, your white world. In order to survive, to thrive, I have to be white for fifty-seven minutes of every hour.

Q: How about the other three minutes?

A: That sir, is when I get to be an Indian, and you have no idea, no concept, no possible way of knowing what happens in those three minutes.

Q: Then tell me that's what I'm here for.

A: Oh, no, no, no. Those three minutes belong to us. They are very secret. You have colonized my land but I am not about to let you colonize my heart and mind. (Alexie 2000, p. 194)

Here, Etta Joseph is visited by the researcher Spencer Cox who want to interview her about pow wow dancing (traditional Native American dance); however, this is not what Etta has in mind. She tells him the story of when she met her first lover, John Wayne. The segment above illustrates the sentiment many indigenous peoples might experience in the meetings with researchers over the years.

I grew up in Oslo (not exactly what one would call a traditional Sámi core area). My father is from Gáŋgaviikka (Gamvik), a costal Sámi village on the northernmost part on mainland on the Norwegian side. In my early twenties, I started reading more about Sámi history and felt a need to connect with Sápmi, Sámi culture, my family and Gáŋgaviikka.

I was unsure about going into these subjects, because I am still figuring out what it is to be a researcher working with indigenous subjects. As a Sámi, I am an insider and I resonate with Etta and her sentiments in the section above. But as a Sámi who grew up outside Sámi core areas, I feel I have to be careful and not act like Spencer, who does research simply for the sake of research, or out of curiosity without realizing that the knowledge that he is seeking is coming from someone, a real

---

1    LA Times, "The artist who made protesters' mirrored shields says the 'struggle porn' media miss point of Standing Rock", http://www.latimes.com/entertainment/arts/miranda/la-et-cam-cannupa-hanska-luger-20170112-story.html.

2    The autonym for a group of culturally related Indigenous Peoples in Canada and the United States that include the Odawa, Ojibwe, Potawatomi, Oji-Cree, Mississaugas, Chippewa, and Algonquin peoples https://www.google.no/search?dcr=0&source=hp&ei=8s6WWtmJHoaLsAGd_7zYDQ&q=Anishinaabe&oq=Anishinaabe&gs_l=psy-ab.3..0l10.906.906.0.1275.3.1.0.0.0.0.158.158.0j1.1.0....0...1c.2.64.psy-ab..2.1.157.0...0.0icn8QLO-AQ.

person with a unique story. Etta knows that she will never get anything back from her voluntary work, and that if she ever got to read the results, she would probably not recognize herself or her culture in it. I think that is why she wants to tell him her own story, to make him see that his ideas about her and her culture are so much more than what he could read about in any book. The knowledge Spencer Cox is seeking is a Western construct and has little to do with the culture that he is researching. It does not contribute to a deeper discussion or understanding in the relation between these two cultures. And it most certainly is not giving anything back to the people he is taking the knowledge from.

I want this article to open up for non-colonial and indigenizing ways of talking about Sámi identity and vuoŋŋalašvuohta, but I also do not want to relate to the "subject" of this article or Marielle, in an exploitative or extractive way in any sense. I keep reminding myself of this because research institutions and educational systems have played an important role in the colonization not only of indigenous lives, but also their minds. It has made indigenous people wary of research, and many have a hard time seeing the "inherent value" of research that many researchers claim. As Etta (A) says to Spencer (Q):

> A: Those books about Indians, those texts you love so much, where do you think they came from?
>
> Q: Well, certainly, all written language have its roots in the oral tradition, but I fail …
>
> A: No, no, no. Those books started with somebody's lie. Then some more lies were piled on top of that, until you had a whole book filled with lies, and then somebody slapped an Edward Curtis photograph on the cover and called it good.
>
> Q: Those books of lies as you call them, are the definitive texts on the Interior Salish.
>
> A: No, there's nothing definitive about them. They're just your oral traditions and they're filled with the same lies, exaggerations, mistakes, and ignorance as our oral traditions. (Alexie 2000, pp. 193–94)

Here, want to open up for a discussion about identity and religion or spirituality in Sámi contexts and to write about it in a way that is on Sámi premises, or on Marielle's and my premises at least. This is why I choose to focus on Sámi and other indigenous stories, to make use of Sámi research, carried out by Sámi researchers and to implement indigenous methodologies, focusing on indigenous experiences. To show why and how I am going to do this the next couple of pages will touch upon the history of Western research and indigenizing and/or decolonizing responses to such research.

For indigenous peoples research, the western paradigm of knowledge and education, what Cecilia Salinas (2020) calls "the pedagogy of detachment", historically has done more damage than it has done them good. Residential schools created a physical break from home, preventing indigenous children from learning their cultural practices and culture-based knowledge. Instead, they were taught Western values, worldviews and practices (Kuokkanen 2000, pp. 412–13).

Salinas (2020) writes about how the pedagogy of detachment has shaped her experiences with educational institutions both in Argentina as a child and in Norway as an adult. The pedagogy of detachment is based on Western values of knowledge and learning, removing the connection between our emotional knowledge and our intellectual knowledge. In school, children learn to categorize their thinking, separating it from their physical, spiritual, emotional, cultural, historical and political knowledge—their situatedness (Salinas 2020).

> By disregarding, negating and exterminating ways of knowing and doing, it has been possible to sustain the idea that the alienation of the mind from the body, and the self from others and from their environments, is natural and desirable. However, this notion has led us away from understanding. ( … ) Knowledge about the world is created and imparted through fragmentation, as the whole is divided into parts. It is in this sense I claim that the pedagogy of detachment is intrinsically colonial. (Salinas 2020, p. 12)

The way a society brings up its children shows us a lot about the worldviews and values of that society. Indigenous experiences with colonialism vary, but they do have many similarities, education and missionization often being the main tools of forced assimilation. In Sápmi, as well as in many other indigenous societies, boarding schools were removing children from their parents, breaking ties between generations and disturbing the most important process of learning the knowledge of their own cultures in children's lives. Victoria Harnesk of the Sámi Association in Stoockholm stated, "We, the Sámi people, have not been subjected to a bloody genocide but of a cultural, 'soft' genocide, based on hidden but effective tools employed by the swedish state to steal our land, water, language, religion, identity, and the possibility to pursue our traditional livelihoods" (Kuhn 2020, p. 8).

The effect of this "cultural, 'soft'" genocide has left its marks on Sámi, our cultures and our health and wellbeing. Suzanne J. Crawford O'Brien emphasizes how Native Women from the South Puget Sound area valued healthy relationships with people, spirits, nature and the dead for the maintenance of a healthy self. The rupture of relationships led to the dissolution of the self (O'Brien 2008, p. 138). In the text called *Restoring Sacred Connection with Native Women in the Inner City*, Alanna Young and Denise Nadeau use the term "psychosocial trauma" to talk about the experiences of many urban Native Americans. They argue that for the women they worked with the relationship with the land and all living beings have been disrupted, causing a spiritual separation. For many, they say, this disconnection led to a feeling of "loss of awareness of connection to both those in their communities and to the land" (O'Brien 2008, p. 121). In essence, the establishment of a self-defined identity through connections with one's community, "the sacred", and the land is more difficult than challenging racism, sexism and colonialism from a place of confidence and "collective power" and to "find one's own truth and resist and challenge imposed structures of thinking and being that have become incorporated in the body" (O'Brien 2008, p. 132).

According to Jens-Ivar Nergård, the most substantial colonization of the Sámi population was not the geographical borders that were drawn, dividing their lands, but rather the offensive against Sámi civilization. While the borderlines were motivated by power interests between nation states, the inner colonization comprise of a more dramatic and thoroughgoing and systematic annexation. Most visible were the attacks on the language. Far less seeable was the offensive against Sámi thinking and world views, beliefs, and cosmology, knowledges and connections to nature (Nergård 2019, p. 21)

The removal of people from their cultures and their lands, however, was based on Darwinistic ideas strongly impacting the scientific writing of the 18th and 19th centuries, reinforcing racial typologies and corresponding behaviors, attributes and capabilities among different populations. Race as an intellectual term and scientific category became a matter of interest, and an important tool in the identity-making of peoples of the Western world, feeding ideas about nations, citizenship, progress, and "otherness" (Alam 2016, p. 80). The cultures of indigenous peoples were seen as lower steps in a series of developmental stages and sociologists would study these societies with the intention of discovering how Western societies had developed. In a way, people now could feel that they were doing the "primitive natives" a favor, taking control over the land they lived on so that it could be used in the most profitable way, teach them how to learn, how to speak, how to dress, how to pray to the right god, and how to feel about themselves.

Decolonization or indigenization aims towards gaining a critical consciousness of oppression, the misrepresentation of history, and discovering our own roles in this unfolding and the degrees to which we have internalized colonialist beliefs and ways of living and challenging the racist, culturalist and dualistic notions prevalent in much of Western ideas and scholarship still guiding many Western and non-Western people's lives. It is also about transforming colonial relations between indigenous and non-indigenous people (Young and Nadeau 2005, p. 10). More specifically, indigenization is about working towards a positive re-building of our communities, it is about empowerment and rejecting victimization and the re-centering of ourselves within indigenous traditions (Stevenson 2000). Here, I will use both terms interchangeably because I do not limit de-colonization to indigenous peoples

only and therefore the term indigenizing will in some cases exclude non-indigenous people from the larger potential that lies in the de-colonizing process.

Jeffrey Sissons, amongst others, comments on the way the nineteenth century's racist ideas have been smuggled into the research of today, and how the ninetieth century idea that "the human species consisted of different races inhabiting different environments and this explained differences in appearance and thought," still is evident in the way humanity today is divided into different cultures, that instead of race "explains" why we think and look the way we do. "Racism now exists as a trace, a ghostly presence that haunts culturalist thought" (Sissons 2005, p. 37).

Kuokkanen reminds us that unlike Western scholars who can ignore this, because outright racist academic approaches has waned somewhat during the last decades, indigenous people cannot remain indifferent since these sentiments still affects us in several ways through degrading and prejudiced beliefs and opinions on indigenous identities, worldviews and cultures (Kuokkanen 2000, p. 413). Within indigenous studies it is thoroughly documented how indigenous knowledge often is reduced to a "subjective", emotional "belief", while Western knowledge is seen to be based on "objective facts" (Sehlin MacNeil and Lawrence 2017, p. 147).

Rauna Kuakkanen underlines the importance of an indigenous paradigm that bring up questions of relevance for indigenous communities and contributes to understanding different ways of knowing and looking at the world (Kuokkanen 2000, p. 414). Here, I raise questions about identity and spirituality, and try to answer them with the help of Marielle and others, in ways that might be illuminating and interesting for some, but also, I hope, helpful or thought provoking for Sámi and other indigenous people that find themselves wondering the same questions, or who might not have thought about these subjects at all.

*1.2. Why Indigenous Methodologies?*

Indigenous research is primarily for indigenous peoples. The main principle of indigenous research is "that ethics and value beliefs that define relations and responsibilities of researchers to the researched should be addressed before ontological and epistemological questions and should drive the research process from formulation of research proposal to dissemination of findings" (Chilisa 2012, p. 20).

To understand the degree to which indigenous people often are connected to their communities, in many ways is essential to the understanding of indigenous ontologies[3] (Chilisa 2012, p. 20). Realising that both Marielle, Stenberg and myself are talking from experiences that relates to a belonging to a Sámi community, whether it is within a Sámi reindeerherding community, a community that is preoccupied with learning and strengthening Sámi joik, or as a Sámi person relating to Sámi communities as a newcomer and a researcher.

Valuing a relational epistemology[4] is an important part of the indigenous research paradigm. That means appreciating the fact that there are many ways of knowing, and that indigenous ways of knowing is based on relationality: the connection and relationships of all things and beings. Looking at a flower in a laboratory is different from studying it where it grows. Indigenous ways of knowing are based on connections (Chilisa 2012, p. 21).

A relational axiology[5] bases itself on the theory of relational accountability, which means that it embraces the fact that all parts of the research process are related and that the researcher is responsible for and obligated to all relations (Chilisa 2012, p. 22) or connections and the responsibility that comes with it. Opening up about personal and intimate Sámi subjects as representatives for Sámi communities to a largely unknown audience is something that Marielle, Stenberg and myself has reflected on.

---

[3]  "Ontology is the body of knowledge that deal with the essential characteristics of what it means to exist" (Chilisa 2012, p. 20).
[4]  "Epistemology inquires into the nature of knowledge and truth" (Chilisa 2012, p. 21).
[5]  "Axiology refers to the analysis of values to better understand their meanings, characteristics, their origins, their purpose, their acceptance as true knowledge, and their influence on people's daily lives" (Chilisa 2012, p. 21).

This means thinking about the ways our words affect our communities. Asking what this research is contributing with to Sámi communities and research is also important.

Relying on Kuokkonen's own definition, the characteristics of an indigenous paradigm are as follows: (1) a social and political agenda aiming at a thorough indigenizing of indigenous societies, (2) a critical view towards Western metaphysical dualism, still shaping most of today's research practices and ways of looking at the world, (3) a holistic approach striving towards a balance between all aspects of life, not separating intellectual, social, political, economic and spiritual forms of human life, and (4) situatedness, an explicit connection to the researcher's own culture, meaning that forms of expression and cultural practice are reflected in the researching process. This is in the use of language, style, structure, methods and assumptions of knowledge and the role of the researcher (Kuokkanen 2000, p. 417). Moreover, this is a direct response to the way research has a habit of separating experience and knowledge from its surroundings and contexts, following the pedagogy of separation. A process that is possible to trace just by looking at the way language is used in research. "Experience becomes data, places and relationships become research sites, and people become focus groups, respondents" (Khan 2016).

The material used here relies on two different interviews with Marielle. The first interview was done the same year we came back from Standing Rock focusing on her experiences there. For the second interview, I asked Marielle if she would like to talk with me about yoik and vuoiŋŋalašvuohta. This interview was done via video call. The interview was informal as Marielle and I know each other and have kept in touch since the journey to Standing Rock. I asked her some specific questions about identity and spirituality, the focus on yoik came through quite organically. As Marielle spoke, the connection that she felt between yoik, vuoiŋŋalašvuohta, the land and identity became apparent. Picking up the threads that she laid out, I followed up with questions relating to her stories, thoughts and explanations. This is a way of doing interviews that lets the interviewee steer the conversation, shifting the power structure away from a classical "researcher—interviewee set up", allowing the conversation to flow more organically and also in a way that makes the interviewees own thoughts and opinions come through. Being a bit unsure about continuing the work of this article because its subject might open up some emotional subjects for some, I asked Marielle what she thought about me planning to write about vuoiŋŋalašvuohta in this way. Her response assured me that as long as I continuously keep asking myself the following questions, I might be on the right track: Why am I doing this? To what cost? In which way? For whose benefit?

In relating to Zintkala Mahpiya Win Blackowl, whom I interviewed for my master's thesis, I informed here about the theme of the article and made sure that she was fine with me bringing up some of her thoughts about the subject. Sten Jörgen Stenberg, whom I did not interview, but whose concerts and talks live streamed via a group called Sámi Sessions on Facebook I have cited here, has seen the article in two versions. His feedback and consent were given. I am grateful for the positive response I got from Stenberg from the very beginning when I approached him regarding this article, which also motivated me to continue this work.

## 2. Processes of Identification

Here, I will go into the subject of indigenous identities, establishing the foundation for the analysis below. Why is a self-defined identity important for indigenous peoples and in what way is it connected to cultural and spiritual practices?

Here, Salinas talk about her own struggle with a destructive system of education and her need to find a way to care for and protect herself from the negative effects it had on her. She says that along with the intellectual capacity she developed at school, she also acquired an inferiority complex and a lack of confidence: "I still have to work hard not to reject the person I look at in the mirror every morning" (Salinas 2020, p. 17).

> I was disconnected to my world, my senses and my capacity to understand phenomena
> from my experiences. Moreover, there was not only an attempt to indoctrinate me through

amnesia, to borrow Vergès' words, "to create a strong disconnect between the child and the world" but also through alienation and devaluation. As such, I had to continue finding ways outside school to develop a perspective of care to protect myself from that devaluation. (Salinas 2020, p. 16)

Salinas' openness provides a view into a rarely expressed (especially within research) experience of the long-term effects of the pedagogy of separation can have on a person. To mend those connections is a way of tending to the wounds of colonialism. These connections lead to a sense of belonging, of safety and a place to ground oneself.

My experience is that identifying as an indigenous person is a process that is not so much about "getting to know yourself," as it is taking back what was taken from you, your family and your ancestors through processes of colonization and assimilation. It is also about holding on to and re-integrating what emerges in this process. It is an ongoing undertaking for many and something that requires a lot of energy and space in one's life. But I would argue that it is worth it in many ways. As Poia Rewi (2011, p. 57) says:

Cultural identity is important for peoples' sense of self and how they relate to others' and contributes to the individual's wellbeing. Identity reassures one's sense of self-worth, confidence, security and belonging. It instills pride. Conversely, to have no culture is to experience a lack of identity. ( . . . ) People without identity are like the tree with no roots to establish itself firmly. It is constantly at disposal of the elements.

Due to the lack of knowledge in Norway about Sámi history and culture, both Norwegians and Sámi people (especially those who have grown up outside Sámi core areas) might have a narrow idea about who Sámi people are. One might argue that the only reason why "indigenous purity" is and has been of interest to settler and post-settler governments because they depend upon varying degrees of biological and cultural authenticity before granting support or recognition to indigenous people. This way of thinking about indigeneity have in themselves become oppressive (Sissons 2005, p. 39). One might ask: "Why should first peoples be expected to have authentic identities while settlers and their descendants remain largely untroubled by their own ill-defined cultural characteristics?" (Sissons 2005, p. 37).

Indigenous peoples are expected by majority-society to fulfill the criteria of what an indigenous person should look like and behave. Movements like the water protectors camp at Standing Rock, however, contribute to the process of turning this around, because it reconnected many young Native American and other indigenous people with their cultures, spiritualities and histories. For young individuals who might have been feeling disconnected from their heritage and culture, seeing other people in similar situations and being able to befriend accepting elders or other knowledge bearers was surely meaningful.

Petrillo and Trejo (2008) write about the internal effects on those living under an oppressive government guided by racist colonialism. When told certain things about oneself by the majority for many generations, it can be hard not to listen, and to keep it at a distance. Sometimes indigenous peoples have to fight to resist these degrading words, and to keep those judgments out of their own heads. In some cases, they turn into self-hatred and internalized oppression and the fight against colonialism has to happen inwardly (Petrillo and Trejo 2008, p. 92). Some would argue that this is the most important fight and that getting rid of internalized colonialist ideas is the key to self-determination. Zintkala Mahpia Win Blackowl said: "The real work is dismantling these structures that live within ourselves".

Indigenous people have been criticized and attacked for living out their lives grounded in their own cultures and belief systems, both through violent processes of colonization and forced assimilation, but also through the idea, prevalent among many researchers, that identity and belonging are constructs and strategies designed solely to gain rights to land or political power. The trouble with concepts such as "indigeneity" and "identity" is the inherent colonialist environment in which they were created.

My point is that although fighting for political power and land rights might have become a part of the struggle for indigenous peoples as an effect of colonization, processes of identification are essential for creating those connections that are so important for our health, mentally and physically. I am trying to convey the inherent emotional need that fuels this process. No matter how complex and intertwined our identities are we are entitled to fully claim our heritage and our histories. Just like everyone else's identities, indigenous identities are varied, and they do not fit any descriptions of what they are "supposed to be."

## 3. VUOIƊƊALAŠVUOUHTA—A Way of Being

When Marielle and I sat down for the second interview, I wanted to ask her about vuoiŋŋalašvuohta and identity in a more direct sense, and what she thinks about the relationship between these two terms.

> Let's say I'm going to sew a pair of *gápmagat* (Sámi shoes made from reindeer hide). Within this knowledge lies so much more than just creating a product. Within that there is a lot of spirituality. Many people think about noaiddit when they hear "Sámi spirituality", something otherworldly, but for me it is about everyday-life. How you act and move in nature. What you think when you're there, how you relate to animals, to the water, the ground, what you do when you prepare materials taken from an animal.

> There is a lot more than you can learn from a book within this (sewing the gápmagat). The practical work is one thing, but it is the thinking behind it and also that it (the knowledge) has passed through all these people for many generations. ( ... ) So what I mean is that when I was younger, I thought I was very distanced from this vuoiŋŋalašvouhta. I thought that this was something that belonged to the old days. ( ... ) Because it was presented in *ungdomsskolen* (junior and senior High School, age 13–15) as a separate subject ( ... ) as something from the past, as old Sámi religion. But vuoiŋŋalašvuohta is a lot more than a religion. And I think that vuoiŋŋalašvuohta, the old Sámi way of thinking, isn't really a religion, it is a way of life, it is a philosophy, it is a way of being[6].

In Sámi traditional knowledge, the boundary between the earthly and the spiritual, the empirically "objective" and the intuitive are porous (Nergård 2019). Lovisa Mienna Sjöberg (2018) too talk about the way categories like religion and spirituality are contested, especially within indigenous contexts. A concept such as spirituality presupposes a separation between the spiritual and something that is not; "nature religion" surmises the same dichotomy. The Western, scientific way of looking at the world, through "facts", "objectivity" and research has declared itself as above the spiritual and the religious. The idea that Western secular societies, distinctly separating the worldly and the religious, are the trailblazers of the right, the real and the true is implicit in this. The term secular refers to the temporal, the worldly and the non-religious. The world of science has taken as a starting point that there is a reality "out there" that can be studied and described. This reality can be reached by studying nature through natural science practices (Sjöberg 2018).

Seen from inside of the Western, science-based, materialistic belief system, religion seems like a pit of superstition and irrationality, representing a step on the latter of development that belongs to the past, even if recent research show that religious people struggle less with anxiety and depression than non-religious people, the numbers of suicide are lower amongst religious people compared to non-religious individuals, and they are less prone to overuse of alcohol or other narcotics than the non-religious part of the population (Sheldrake 2017, pp. 20, 22). For those who believe in the materialistic nature theory where everything is mechanical, causation based and all emotion an effect of endorphins and hormonal changes, choosing a life "without meaning" can seem like an heroic

---

[6]    All of Marielle's quotes are translated from Norwegian by me.

act, a brave oath of allegiance to the "objective truth". However, the philosophical materialism is not the truth itself, it is a worldview, a belief system. Even if it has many devotees, the belief in it is not a question of intellectual, logical necessity, but one of ideology or personal or cultural habits (Sheldrake 2017, p. 70). The inherent belief in materialistic nature theory is mirrored in the division between the religious and the non-religious. As Sjöberg says:

> The term faith and faiths are associated with a secular worldview. A worldview where religion and religious practice, especially christianity, has been studied as a phenomenon belonging to the private sphere. This leads us away from the practical and material sides of religious practice. (Sjöberg 2018, p. 19)

Sjöberg reminds us that realities are not explained by practices and beliefs but are instead produced in them. Realities are produced and have a life through their relations (Sjöberg 2018). Yoik and yoiking can be seen as one of these practices, connecting the listener or the juoigi to their surroundings or inner worlds in ways that creates and affirms lived realities also on a spiritual level.

The fact that yoik can be seen as a spiritual practice might not sit well with everyone. Marielle, however, sees yoik as vuoiŋŋalaš, and she is not the only juoigi I have heard talking about the healing power of it and of the connections that are made through it. Here, Marielle talks about what she sees as the relationship between *luohti* (yoiking song, subject) and vuoiŋŋalašvuohta:

> I have always known that of course there is a link both to the past, future and now in luohti, and that there is a strong vuoiŋŋalašvuohta in there. I learned very early, it was a juoigi who told me when I was young, that these *luođit* (plural form of luohti) that were used in ceremonies, those that we have had through history, and that no longer exist in that form, these juoigis said that there have been luođit that have been ceremonial, that have been in use only in these ceremonies that are a bit different than those that are yoiked today.

> So when I was young I thought that these vuoiŋŋalaš (spiritual) luođit were gone. ( ... ) As I became a traditional juoigi myself, I was joiking for many years, learning more and more, until I felt I had the basis to call myself a juoigi. But along the way I learned that luohti is vuoiŋŋalašvuohta, but on many different levels. It is between humans, it is in communication between people, it is in the way you perceive the world. So it is clear that it is like you write in your thesis about Standing Rock, of course luohti is essential to identity and vuoiŋŋalašvuohta.

Before we left for Standing Rock, Marielle talked about how she hoped to be inspired by the indigenous people she would meet there to explore Sámi vuoiŋŋalašvuohta further. In the first interview we did together the same year we came back from Standing Rock, I took the opportunity to ask her if she felt that her time at Standing Rock inspired her to dig deeper into these subjects.

> There is a lot in Sámi spirituality that is unsaid. Sámi spirituality lies in the way we are and what we think. So we don't speak very loudly about it, but it's there. ( ... ) And the camp, it was very spiritual, and all of it was very familiar, the not so familiar part was the vocal aspect of it. That so much was said aloud, like, "Now we're going to have this ceremony." But I think that if its (vuoiŋŋalašvuohta) to survive we'll have to talk about it, because it will disappear if we're not conscious of it. I feel that we're the middle generation who were taught by the elders, who in turn practice it, but never talk about it. We have learned by observing, while not actually practicing it ourselves, so how will the kids learn, right? Then we will actually have to talk about it and do it consciously.

Similarly to Marielle Lovisa, Mienna Sjöberg stresses the need for a verbalization of Sámi knowledge, translated from Swedish by me:

> Those arenas where knowledge, also about spirituality, earlier were transmitted are disappearing and replaced by new ones. In these situations there arises a need to verbalize

and transform this knowledge so that it becomes transferable to other arenas and the coming generations. Maybe it becomes extra vulnerable when the everyday use of these practices aren't verbalized and formalized, since the everyday is also what is changing so quickly. (Sjöberg 2018, p. 8)

During a conversation with a *guvllár* (someone that can heal and help you solve problems, communicate with the being of the ground and/or find lost objects) in Guovdageaidnu, she told me how her mother had told her to be careful and keep quiet about her powers because they used to burn people like them[7]. It was surprising to me because to me the witch hunting days seem to be a long time ago, but for her it is still a living memory. It made me realize how hard the struggle to reinforce Sámi spirituality still is today and how present the memories of condemnation against Sámi rituals and spirituality can be, especially in the Sámi core areas, where the connection to elders, who have these experiences even fresher in mind, is strong. This might be part of the reason why the verbalization of Sámi spiritual knowledge has been muted and led to some additional challenges for people like Marielle who wants to strengthen the traditional Sámi spirituality.

Yoiking is also a part of this verbalization or vocalization process for many, connecting Sámi individuals to their communities and lands. The relationship is not as straight forward for many Sámi as one could wish, however. It is also steeped in a colonial soup of missionization and forced assimilation. In the next section, I talk about the changing role of yoik in Sámi communities, and the way that yoik in itself can serve as a means to heal and gain new strength in a world where Sámi lives and livelihoods are threatened by colonial and capitalist forces.

## 4. Strengthening Connections

Yoik in itself is and has been a way to communicate and create bonds between people, to the extent that, in some cases, according to written sources, people have been using it instead of speech. In the 1800s, Petrus Læstadius, a Swedish priest and missionary, spoke of how yoik and speech were blended.

As the conversation becomes more lively, the prose ceases and 'one begins to talk through song. One's feelings are expressed in song, people start to hug each other, and one talks and responds to each other with song'. (Læstadius quoted by Graff 2014, p. 71)

Another story about yoik as means of direct communication is from a text by Ola Graff, a Norwegian researcher specializing in yoiking. Here, he talks about a Swedish teacher who knew two friends whom at times changed from talking to yoiking when "some strange mood came over them." Graff quotes the teacher saying: "Now and then it might not be possible to resolve minor conflicts with ordinary prose but 'with ascending and descending tones of Sámi rhythm, they 'overwhelmed' each other with what was in their hearts'" (Graff 2014, p. 71). More than talking about the functions of yoiking, these stories convey its strong presence in Sámi traditions. The process of Norwegianization was an attempt at exterminating yoik, amongst other Sámi traditions.

Some Sámi people today have conflicting feelings about yoiking. An example is an experience in Guovdageaidnu, where I studied North Sámi at the Sámi University of Applied Sciences with a group of around 20 Sámi students of various ages, many of them around my age at the time, 26, or a bit younger. The class was asked by a teacher to yoik one of our teacher's yoiks, as a sign of gratitude at the end of our first term. However, many of our classmates almost did not make any sound during practice. One of the students spoke up, saying that people might be uncomfortable with yoiking, because they do not feel they have the skills and experience to be able to do it properly, at least not in

---

7 Later she said that she wanted me to specify that the noaidis where the ones who got burned, but that a guvllár had to be careful not to be noticed for their powers so they would be burned too. In her view only noaidis were the ones who had the powers or the will to affect others negatively. This is not an opinion shared by all Sámi.

front of others. When we changed the yoiking sounds into more Western sounding singing sounds, people seemed to feel more comfortable actually singing the melody out loud.

I could empathize with the feeling of my classmates, still struggling to break through these barriers inside that we in some way have been conditioned to put up. Being the daughter of a *juoigi*, I have grown up with luothi. I have quite recently started to learn yoiking, and have found it challenging at times, the sounds and melodies hard to master, and many times having the feeling that it is too late for me. But I think this is a feeling that is not so much grounded in the bodily capacity but in the psyche.

A part of it for many might be the feelings described by Hämäläinen et al.'s text about the use of yoik in dementia care in Northern Norway: "On the one hand, one might long for the language, the yoik, the acknowledgement of who one is. On the other, one may bear a general sense of shame for the very same things, transferred through generations of Sami people who were oppressed and devalued" (Hämäläinen et al. 2020, p. 33). As Petrillo and Trejo (2008) write, the narrative written by majority society has an immense effect on minorities, and in this case the devaluation of yoik and Sámi culture has left its mark on many. This is where the internalized oppression has to be recognized and dealt with.

What might seem like a contradiction at first glance is the presence of the ghost of oppressive authenticity in this situation. Sámi people from outside Sámi core areas might feel that they do not have the skills or even the permission to yoik. Because there is a lack of realistic and "non-authentic" Sámi role models portrayed in popular media, representing the reality of many Sámi (those who do not speak any Sámi language, those that never have yoiked or who grew up in a city), some might feel like they do not fit into the image of who a "real" Sámi is. This can stop us mentally and physically from taking part in activities like yoiking. The fact that the Norwegian reality shows about yoik mostly portrayed non-Sámi learning yoik from competent Sámi *juoigit* (or an already competent juoigi preparing with other juoigis to perform), instead of including Sámi who do not know how to yoik and who want to learn, also attests to the unwillingness to go into stories about "non-authentic" Sámi. This would probably have created some discomfort for the Norwegian audience but would have been a great opportunity to talk about why so many Sámi have been disconnected from this part of their culture. This part of Norway's history is unknown to many Norwegians.

Yoiking today is something that many might see on television or at concerts, instead of at home in a non-performative setting. It seems like yoik has been moved out of the private sphere and into the public one. Yoik is more and more being appreciated in a performative setting, through recordings, concerts and TV-shows than in the private, intimate and practical way. Nils Oskal (2014), a Sámi professor in philosophy, talks about the Sámi *náhppi,* and its use in Sámi reindeer herding communities. The náhppi was used by the women to milk their reindeer. By the end of the 1940s, the reindeer industry and the communities overall underwent a series of changes, which resulted, among other things, in the end of reindeer dairy production and the economy began to focus solely on meat production. This change happened at the same time as the institutionalization of *doudji*, Sámi handicraft. And, as the náhppi gradually lost its practical function during the 1950s, it gained a new life in the duodji-tradition as an object of traditional Sámi handicraft (Oskal 2014, p. 88).

The tendencies that might point towards a changing role of yoik in the daily life of Sámi people can be compared to the náhppi's situation. Yoik seems to have been moving out of the practical and private sphere into a more institutionalized and formal one. This is a part of a longer process, starting according to many with Nils-Aslak Valkeapää, a multitalented visual artist, juoigi and poet, who became one of the first to merge yoik and rock with great success during the late 1960s. Through him, yoik became a strong symbol of Sáminess. Yoik seemed to be ready for stages and recordings, it became a much awaited and needed source of pride and created bonds between many Sámi people (Angel 2015). But what happens to yoik when it is brought up on stage? Ánde Somby, a *juoigi*, reflects:

> The feature of dialogue becomes a challenge when the yoik came on stage, it became a monologue. That was maybe the part of the negotiation that was central in the Davvi Šuvva[8] time, the yoik was now on stage, but how should it be? What kind of form should it have? ([Angel 2009](), p. 96)

Although I am still seeing many examples of Sámi taking yoik back into their lives, it might seem like the role of yoik on a larger scale has shifted. This is of course a very natural process. Practices change and peoples interests and priorities will always fluctuate. Yet it is also a fact that the reason for this changed relationship is colonialism and the strict assimilation politics, specifically targeting yoik and Sámi spiritual practices.

For Marielle who has been encouraging people around her to yoik, both through YouTube videos and in real life[9] through courses and talks (and through mentoring a Norwegian participant taking part in the TV-show Muitte Mu), there is more than yoik in itself at stake. For her, it is also about maintaining and strengthening the core of Sámi culture and spirituality. And while Marielle is working for yoik to become a larger part of people's lives again, yoiking has also become something that some Sámi today feel a certain distance towards. This is what Oskal says about the institutionalization of duodji and its effects on the role of the náhppi:

> The organization Sámi Ätnam appointed a dedicated duodji consultant. This was partly due to a desire to allow those making duodji to earn a living from it. ( … ) Courses on judging duodji were also held. I find this very interesting. It is something new, and it is the starting point for a new set of autonomous criteria for evaluating the náhppi.

I myself have gone to a yoiking course at the musical conservatory in Tromsø, where our end exam was to perform a yoik that we had caught[10] ourselves, to which we received a grade based on what is considered a good, traditional yoik and what is not. There have also been two different TV shows ("Muite Mu" and "Stjernekamp") where the participants, most of them with no experience in yoik, have had to learn yoik over a period, guided by competent yoikers and in the end give a performance that was, in some instances, to be evaluated by well-known yoikers from the Sámi community.

Oskal describes the similarities between the history of the náhppi and the liberation of art. According to Max Weber and Jürgen Habermas, he says, "autonomous art emerges alongside institutions of art critique as a result of division forming within society" ([Oskal 2014](), p. 86). From this division, the institution of art, along with a unique expert evaluation system is established. In the same way, the yoik courses and TV shows are creating opportunities for people to learn about yoik and yoiking. One might argue that a tendency toward a separated institutionalization of yoik can contribute to further alienation for many Sámi people. And the risk is, as Kuakkanen says, that institutionalization has a binding effect, facilitating structures of power and authority. However, it needs to be said that, for many Sámi people, one of the few opportunities to learn yoiking today might be through these channels. But the question is, inspired by Oskal's analysis of the evolving náhppi, can the yoik be taken so far away from its practical and communicative context that it loses its original meaning or purpose, its ability to relate and communicate with people, and with that its connection to the spiritual?

Marielle, who clearly has given a lot of thought to the subject of Sámi people's mixed feelings about yoiking themselves, also has a more positive view on everyday yoik and its current role in Sámi cultures.

---

8   Dávvi Šuvva was a Sámi festival held for indigenous peoples for the first time in 1979. It took place on Kaarevaara, on a hill west of Gáresavvon/Karesuando on the Swedish side, but there were also events taking place on the Finnish side Kaaresuvanto ([Angel 2015]()).
9   [https://www.youtube.com/watch?v=H2LhBAi-Q8I]().
10  The way a yoik comes about in our world is by being caught by someone, "it comes from the big everything and will show itself to you. The idea is that the yoik is out there, and will manifest itself to the yoiker." Torgeir Vassvik (2019, private communication), juoigi.

> (W)hen it comes to luohti and yoiking I don't see it as totally negative that people have this barrier. Because that shows that you have a very personal relation with it, that you have some kind of feeling of ownership nevertheless, and a respect for it.

For Marielle, the ambiguos relationship to yoik and yoiking that some might experience can be a sign of regard that is there despite norwegianization and missionization. It is a relationship after all, and thus it is not a broken tradition. It is a relationship that can be strengthened. Marielle has been encouraging people around her to yoik, both through YouTube videos and in real life[11].

> So I am working with yoiking-courses, but I can't say I'm a yoiking teacher, because no one can teach anyone to yoik, it is a personal journey that everyone has to work on themselves. But what I can do is to be a kind of a motivator. ( ... ) But as I said this is a journey and you're not a juoigi within a course or two. It's a long journey. But everyone is allowed to be a beginner. And to work with their voice and their identity. I wish that all Sápmelaččat (Sámi people) would work with this (yoiking) or at least establish a relationship with it. «Do want to be a juoigi or not?» At least think about it. And «why not?», or «why do you want to?» and «how?». And don't just let it slip away without thinking about it. I don't want that.

For Marielle, yoiking is also a way of connecting to the land and to nature, and she says that she knows what a disconnection from place and nature feels like.

> It is hard in those areas where the tradition is totally broken. Because luohti and identity is strongly linked, because of that special juoiganmalle (yoiking dialect), that dialect that belongs to you belong to the place you are from and the family. But I understand that it is hurtful. I have those feelings myself because I never got to hear luohti from my mother's area which is totally Norwegianized. And I feel that that it is difficult for me to fully connect to those waters and that area where my mother was from because I haven't heard those luohti, I will never learn them.

Nevertheless, she believes that a relationship to ones homeplace and the land and waters can be restored through yoik.

> Why it (luohti) is so connected to the place you're from and your identity is because luohti isn't just a song about that specific place, it is the melody *of* that place. The best way, I think, is if you can hear an older luohti, but if it doesn't exist its good to hear a new one too. If you just catched or put a yoik, that is also good. So I mean that you can revitalize that part of the culture in those places where luohti is gone. But it takes more than just one person, it takes more people to work on it. Becoming yoikers and working to find the area's dialect.

In our most recent interview Marielle shared her experiences as a performing yoik artist, accentuating that for her there is a clear difference between performative and everyday yoik: "I am a Sámi artist myself, a stage yoiker, but it has been and still is very important to me that I continuously define to myself what I'm actually doing". While Marielle distinguishes between performative and private yoik, Sten Jörgen Stenberg shares other perspectives regarding the effects of yoik, also in a performative setting, below. Marielle is also clear on the fact that, even though she has been teaching yoik to non-Sámi, she believes they can never call themselves a juoigi. And that is just because of the connections she talks about above. Commenting on a near-finished version of this article she said, or rather, wrote:

---

[11]　https://www.youtube.com/watch?v=H2LhBAi-Q8I.

I would like you to add, where it says that I urge people to yoik more, that this is about our people. I also have courses for non-sámi, to spread knowledge about our culture. It is the practitioners of a culture that holds the right to develop their own culture in the directions they prefer, and it should not be defined by people from outside that culture and society. That is why I think that non-Sámi can't be yoikers and use it to their own preference. Do you understand what I mean? During these courses I talk about why and how it is so connected to Sámi identity (as you already mention, it has to do with ancestors, connections to place and so on). It is yoik as a language/communication that is threatened, yoik on stage and the likes of it is a one way communication and mostly entertainment.

For many indigenous people re-claiming identity has to do with a return to tradition (O'Brien 2008, p. 9). "Traditions", as James Clifford observes, "articulate—selectively remember and connect—past and present" (Clifford 2013, p. 57). Traditions are constantly in the making, engaging with the now, revisiting the past, and exchanging ideas, concepts and motivations with its surroundings. In connection with indigenous health and healing, traditions are often seen as the right path toward living a healthy life physically and mentally (O'Brien 2008, p. 11). Young and Nadeau write that "The ability to survive and even thrive in the face of adversity" has been linked with tapping into our own natural spiritual resilience," and that "cultural identity is a primary source of strength and spirituality is a core aspect that contributes to this cultural resilience" (Young and Nadeau 2005, p. 2). Furthermore, they write:

> The many aspects of colonialism that attacked religious identity, many of which continue today in different forms, also eroded communities' sense of their spiritual identity. For many this spiritual dispossession was and is expected in loss of awareness of connection to both those in their community, and to the land." (Young and Nadeau 2005, p. 4)

As the movement in Standing Rock also points to, it is not only missionization that is creating disconnection; extractive projects on our lands are doing the same. The fight for undisturbed lands and clean water is taking place in Sápmi too. This is a story from Sten Jörgen Stenberg, a reindeer herder from Málage (Ume Sámi) or Maalege (South Sámi) (Malå on the Swedish side of Sápmi). During the heaviest phases of the COVID-19 quarantine in Scandinavia, Stenberg live streamed a mini concert via Facebook to the group Sámi Sessions from his kitchen table. He starts the following part of the concert by showing the area where his reindeer graze. The map is filled with colorful shapes, dots and lines, all representing different activities and settlements in the area that gradually have taken over the land that used to belong to the reindeer. As a way of telling the backstory to the yoik *Losses Beaivi* (a difficult day), he talks about the ways the land itself has become separated from its vuoiŋŋalašvuohta, the beings of the land, and how the local reindeer herders are being separated from traditional Sámi ways of living, spirituality and life views and with that their sense of belonging and identity.

> You first have to understand this picture to understand the pressure that the reindeer husbandry is under. And this whole situation that led to this yoik Losses Beaivi. Because we human beings are changing the grazing patterns and the grazing peace is changed and disturbed. But the human grazing peace or our soul peace is also disturbed in a very deep sense. And that is hard to explain in a society where all these points (on the map) symbolizes development in the Swedish society. And the fact that we are changing, and reindeer herders ways of thinking are changing, and amongst the Sámi, so that we are almost giving up without noticing. And we are talking about this in a very western way when we are explaining how the pastures are disturbed[12].

---

[12]    All quotes from Stenberg is translated from Swedish by me.

But an old, old reindeer herder who is dead now was there when the first windmills came. We had a heart to heart conversation and he cried and was very sorrowful because the spirituality of the place disappeared, as well as the beings of the ground. And he felt burdened by this because it felt like he had betrayed; even if it was not his responsibility to defend it all, he felt as if it had been his life mission.

> And when you lose the spirituality it is easy to say that reindeer husbandry is just an industry. That we have to look at it in an enterprise wise type of way. And there are people within reindeer husbandry who think about it in this way. ( . . . ) But if you lose your own attachment to the land via spirituality, then we lose ourselves. And it was this sadness that led to this yoik. And I felt very weighed down at that time. And the perspective shifts a little when you . . . In my case, I carried a colleague to the grave because he shot himself. And one does know others who have been in the same situation. And many who have been feeling very, very low. And that's when the perspective changes and you see what's important and what isn't. And this yoik, conveying all this pain, it was needed just then. Simply to unload those emotions[13].

Here, Stenberg talks about two subjects central to this text that I will bring forward: the relationship to land and its beings and his healing use of yoik in this case, and the way that these are connected. The deep sense of loss that comes with the disconnection from the land and vuoiŋŋalašvouhta there rises a need to connect again. It seems like yoiking, for Stenberg, facilitates both an outlet for these emotions and a way of finding one's way back to "what's important".

Marielle has an interest in both of these aspects of yoik. At Standing Rock, yoik was valuable for her as a means to connect to the host, the Lakota people, and to create a relation so that, in Marielle's own words, the Sámi group could: "Learn more from the Lakota about how we could help them protect their land and their water." This is Marielle's account of their first visit to Standing Rock:

> When we came there, me and Inger, we were the first Sámi people. We wore our *gáktis* (Sámi traditional clothing) and we were just being ourselves, but we were not "approved", in quotation marks, as indigenous people until we had yoiked. Then we were. And we got asked a lot of questions ( . . . ) people were interested in who we were and where we came from. And we got to share and talk more about that and I think we got to learn other things than we would have if we were non-indigenous. So in that way it was a very good thing to yoik.

> Also it was a very special thing for a yoiking heart to hear people singing and singing all the time. And that we could fall asleep to the drumming and singing, and waking up to it and to people walking past the *lávvu* (Sámi traditional tent) singing. It was in the people's pulse that September. Everything was song. ( . . . ) So that was proof for me that the traditional music is so crucial when it comes to giving people a sense of community and a common ground. And that also confirms the sorrow I feel for the ongoing disappearance of yoik in the daily life (of Sámi people), because I know it is so important for us as a people and our identity and spirituality. So that is something that we still have. Yoik and traditional singing is a part of the strongest and the most spiritual we have left, and our language of course. So that (the music) was a very beautiful thing and something that made me and Inger feel very much at home.

Sometimes it can be hard to convey the multiple feelings, stories and associations that are necessary for people to meet and understand each other on an emotional level. Through yoik, they managed to convey, without words, something that might have been hard to explain in another way. They earned

---

13 https://www.facebook.com/jorgen.stenberg.5/videos/3089166351122068/.

the willingness of these people to communicate[14] and let Marielle and the rest of the Sámi group know how they could help protect the water and the land in the best way. Marielle's ideas about yoik being important for peoples feeling of community and belonging was confirmed at Standing Rock, validating the sadness she has been feeling about the disappearance of yoik in people's everyday lives.

Oskal's account of the changing role of the náhppi mirrors a possible development in the role of yoik in the daily lives of Sámi. Marielle and Stenberg's stories show the emotional commitment to yoik that many still have and attests to the willingness to strengthen and develop the connections that already is present in yoik and yoiking.

## 5. Situating Our Bodies

It has been established that the relationship to land is often considered to be central to the identity and with that the wellbeing and survival of indigenous peoples and communities (I would argue that this is true for all people, however). Rauna Kuokkanen says that "Regardless the region, indigenous people commonly describe self-determination as a relation with the land" (Kuokkanen 2019, p. 40).

Jens-Ivar Nergård talk about Maurice Merleau-Ponty's ideas about the body not being an object to our consciousness, but itself a form of consciousness. In this way of thinking, the body is not an object, as it is presented when one believes that body and soul is separate, but an active and experiencing subject. Merleau-Ponty talks about the living and performing body, emphasizing how we understand through and think with our bodies. And how we are connected to our environment through our bodily activities, localizing our consciousness about the world. Our bodily actions are centered in our consciousness and unite body with understanding. Understanding and thinking are connected to the activities of our bodies in motion or stillness and our activities "install themselves" in our bodies (Kuokkanen 2019, p. 65).

Nadeau and Young came together through an interest in the body as a site of decolonization. They see the process of decolonizing the body as a journey of discovering its voice, innate wisdom and goodness, and to re-connect with land, culture and community (Nadeau and Young 2008, p. 117). Reestablishing a connection with one's body, or a connection between the body and spirit as Nadeau and Young say, is crucial both for healing and decolonizing and they see healing and decolonizing as two parts of the same process (Nadeau and Young 2008). Taking responsibility for our own health and choosing to be well through traditional medicine and lifestyle is also to take a stance against assimilation and colonial control (O'Brien 2008, p. 9).

Young and Nadeau have more specifically used singing as way of "re-inhabiting one's body", "recovering one's voice" and building a "cultural self-esteem" (Young and Nadeau 2005; Nadeau and Young 2008). More specifically, Nadeau and Young have developed a form of "cleansing ritual" addressing internalized oppression where they "cast off" "negative feelings or self-concepts that have been generated through cultural stereotyping and injustice—that live in the body" (Young and Nadeau 2005, p. 8). Similarly to Marielle who also uses yoik to connect and strengthen relations to one's own identity and spirituality, Nadeau and Young use the voice as a tool when working with the women in their courses.

> We know how violent internalized racism is. So instead of having to introduce an intellectual analysis of the entire colonial historical dynamic, we are able to sing a song publicly

---

[14] It should be mentioned that this is not the first time yoiking has created a room for dialogue between the Sámi and other indigenous peoples. When all the indigenous representatives met for the first time in United Nations Permanent Forum on Indigenous Issues, in 1975, the Sámi and their "indigenousness" was questioned. The South-American Indigenous representatives "did not want to acknowledge "those rich white Europeans" (Sámi people) as indigenous and protested against the presence of the Sámi people. Anthropologist Helge Kleivan had to "emergency-lecture" on the history of Sámi people in Spanish. However, the one who eventually managed to convince them was Áillohaš, who entered the stage and yoiked for the assemblage—skepticism and distrust was yoiked away and the conference could begin." (Harald Gaski cited in Angel 2015, translated by me).

and voice our sacred power in song as a way of restoring pride in where we come from. (Young and Nadeau 2005, p. 6)

Nadeau and Young are clear on the fact that these methods of connecting to one's body, one's community and nature are effective ways of healing. They use what in a Western context might be called "untraditional" methods, but song and movement, localizing and working with trauma in the body are all elements that are slowly taken back also within Western traditions of health work. These methods of healing, connection and wellbeing have been a part of human life for as long as we know and yoik is one of them (Sexton and Stabbursvik 2010; Hämäläinen et al. 2017).

In his second yoik concert from his kitchen, Sten Jörgen Stenberg talks about his experience with yoik in dealing with losing his father and uncle whom he was very close to. They had been working with reindeer since they were young, his uncle Bertil started in 1930, and through them, he says, he got access to that time which was very important to him.

> They both died at the same time and it came as a shock. And sorrow comes in many forms. And for me, it was like going into this state of sickness, as if something was happening inside my head, somehow. I wasn't myself for a long, long time. I couldn't get out of that state of sadness. It got ahold of my body and soul and gripped it firmly, and it locked me down. And then I went down into the garden I have that is behind my house, during the summer or fall. I remember that moment exactly. So I walked, and wherever one looked, one only saw the memories. That was how it was everywhere, one lived only in the past. One is only sorrow. And that's when I decided to try and yoik. And just with the first notes the heaviness let go. That which had locked my mind and paralyzed me. And already with the two first notes, something happened. And so it was that I started using this, so I yoiked myself well, or to the degree that I could start working again. This yoik became the antidote, and I yoiked it only when I was alone, for several years. No one else got to hear that yoik. It was so incredibly private. I tried performing it a couple of times, but the tears would come. It was too emotional. But when some years had passed it got the opposite way around, I could almost not walk up on a stage without performing it. And that is partly because I think its beautiful, but it is also something I want to share with my environment, because I, and this might be pretentious of me, but if it had that effect on me it might have that effect on others[15].

Stenberg recovered through yoik, and he feels that this effect can be passed on through yoiking for others. This theory is supported by a recent study called "The art of yoik in care: Sami caregivers' experiences in dementia care in Northern Norway" by Soile Hämäläinen et al., concluding that "The participants expressed that the bodily impact of yoik is clearly visible in elderly people living with dementia."[16] (Hämäläinen et al. 2020: "Yoik enlivens and brings to life" section, para. 1) and that "Yoik may manifest and enhance connectedness to oneself, to the natural environment and to the community" (Hämäläinen et al. 2020: "sammendrag" section, para. 4).

At Standing Rock, there was one particular yoik that was more present than others and it might have been given a bit more attention because of its history that is short compared to other yoiks, but that already had gained a powerful and extensive web of connections. The yoik *Gulahallat Eatnamiin*, or "we speak Earth", was caught by Marielle and became the bearer of one of the central messages from the Sámi activists and artivists who went to Paris during the United Nations Climate Change conference in 2015. The group of Sámi played a significant role, emphasizing the indigenous presence in the protests taking place during the meeting. Gulahallat eatnamiin was yoiked here, both by Sámi and others. "We speak earth", as it was called in English, conveys the understanding of Earth's language. Gulahallat Eatnamin, however, in North Sámi, translates more directly as *communicating*

---

15 https://www.facebook.com/jorgen.stenberg.5/videos/3040445852660785/.
16 All quotes by Hämäläinen et al. (2020) translated from Norwegian by me.

with the earth. And as the Sámi artivist Jenni Laiti says, talking about the meaning of this yoik, "It takes two to communicate", establishing a relationship with the Earth, not solely interpreting its language (Sandström 2017, p. 78). In Laiti's opinion, this perspective is one that is part of the core in Sámi culture. The person yoiking a place, a river, a human, is in close relationship with these. You are not just beside them, you *are* the water of that river, the fish living in it and the people belonging to that place (Stoor 2017, p. 202).

Marielle and the other yoikers in the group at Standing Rock yoiked Gulahallat Eatnamin as a way to present themselves, but also as a way to connect with the land, with their hosts and their struggles.

## 6. Decolonization Is a Process That Demands a Reconnection with Each Other and/including the Earth

Building a relationship to our heritage, ourselves, the land on which we live and/or the land we or our ancestors came from is important for everyone. In Sámi traditions, these relations to land and community are strong, but, for many Sámi, they have not been. Today, indigenous lands, ways of life and cultures are constantly under threat. At Standing Rock, Marielle became more conscious of the importance of yoik for Sámi people, and of a more vocal spiritual tradition. Like Salinas, we can center on the challenge of detachment. The project of identification is not only for those with indigenous or minority backgrounds, everyone is entitled to a sense of belonging and feeling of connection to one's surroundings, community and history. The movement at Standing Rock was a fight against disconnection—the same sense of disconnection that facilitates all extractive projects and destructions of land and waters in the name of "development". Letting go of the ties to lands, our bodies, our histories, our water, our communities, our songs, our ancestors, our spirituality, and the beings of the ground is a process that is hurtful for everyone. However, indigenous peoples today are often the last bastions of these experiences and can tell stories today about the pain of experiencing these disconnections and the fights that are taken to re-connect, re- integrate and heal, while standing up against further extractive projects on indigenous lands and waters.

The role of yoik within Sámi communities have gone through many changes. Most recently, a move towards a more performative setting is prevalent. Although this is a change that is also deeply connected to the process of assimilation and colonization of Sámi communities. A push toward a more open and proud yoiking culture in the 1960s, led by Nils-Aslak Valkeapää, took the yoik up on stages. It created a new context for yoik and reached a larger audience. Today, yoik has spread even further than ever and is embraced by popular culture and reality TV. It is worth asking how this development is affecting yoik on a deeper level. Maybe as a response to these more recent changes people like Marielle and Stenberg are sharing their knowledge via social media and through concerts and workshops. They have similar thoughts about yoik and its capacities, while they have different views about yoik in a performative setting. Marielle focusing on performative yoik purely as entertainment and Stenberg on the possibility of a healing aspect also within this context.

As we have seen, yoik has been an important part of both Marielle and Sten Jörgen Stenberg's lives. According to Marielle and Stenberg, yoik has the potential to connect people with their homelands, with a vuoiŋŋalašvuohta, and with themselves. This is supported by Hämäläinen et al. and Young and Nadeau who also talk about the importance of yoik and song to the health and wellbeing of people who have experienced colonialism on their bodies.

Writing from Sámi perspectives has been emphasized in this text, making use of Sámi research and resources and opening up for a discussion on Sámi premises. In this way, it is easier to produce research that gives back to the communities it builds on, and perhaps contributing to a constructive debate around vuoiŋŋalašvuohta, identity and yoik.

**Funding:** This research received no external funding.

**Conflicts of Interest:** The author declares no conflict of interest.

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
