# Peer review of "VUOIŊŊALAŠVUOHTA—Sámi Spirituality, Yoik and Its Relations"

_religions, doi:10.3390/rel11100512_

Round 1

Reviewer 1 Report

Slight corrections include:

'it's' in title (line 3) needs to be 'its'

line 12, remove 'from'

lines 111-112 (grammar)

134 gr

274 sp

365 gr

447 sp

(remarks about translator need to be in footnotes, e.g. l 236)

Author Response

'it's' in title (line 3) needs to be 'its'

Done

line 12, remove 'from'

Done

lines 111-112 (grammar)

Personal and communal identity stand as grounding pillars in the work of indigenous activists, community workers, politics, artists and researchers all over the world.

134 gr

I grew up in Oslo, not exactly what one would call a traditional Sámi core area.

274 sp

Let’s say I’m going to sew a pair of gápmagat (Sámi shoes made from reindeer hide).

365 gr

In these situations there arises a need to verbalize and transform this knowledge so that it becomes transferable to other arenas and the coming generations.

447 sp

      Could not find this

(remarks about translator need to be in footnotes, e.g. l 236)

Done

Reviewer 2 Report

The article should be improved using classical IMRAD form which is basic for all of quality scientific studies. This form will help the authors to focus on needed parts of their article, that are in the submitted form absent.

I do recommend focusing on one particular aspect of the topic and not trying to reduce all content of the master´s thesis to one article. 

Avoid popular, and use scientific language with its strict terminology.

Try to put all the main findings in an compact conclusion. Avoid description, and be exact. 

Author Response

The article should be improved using classical IMRAD form which is basic for all of quality scientific studies. This form will help the authors to focus on needed parts of their article, that are in the submitted form absent.

Thank you, this was helpful.

I do recommend focusing on one particular aspect of the topic and not trying to reduce all content of the master´s thesis to one article.

I have now reduced, at lest shortened some of the the topics. I think some of the callenge for me was to show how they all fit together. I hope this seems clearer after this revision.

Avoid popular, and use scientific language with its strict terminology.

I have tweaked the language somewhat, however I wish to write in a style that is understandable also for people outside academia.

Try to put all the main findings in an compact conclusion. Avoid description, and be exact.

I hope the conclusion is clearer and more informative now.

Reviewer 3 Report

This was an interesting paper to read. It revolves around themes as identity, yoik and spirituality.  The main material is contained by an interview with Marielle Gaup Beaska,  and use of Sámi research. The strongest aspect of the paper is its originality and the fact that it makes use of Sámi research, which is important in indigenous methodology but seldom done in practice. This paper carries a lot of information. With some sorting and a better structure it will contribute to the discourses about Sámi spirituality and yoik today, a topic that is less discussed from a Sámi point of view.

General comments:

Make sure to follow up theories and statements throughout the text.

Make sure the sections connect with each other.

Narrow down to a few traces, save the rest for coming papers. I suggest that the authors own struggling with identity could be put in another paper, since this would be relatively easy done. Interesting, but enough for a new paper.

It has some structural challenges. Because of the importance of the paper I have chosen to comment it quite thoroughly, and I hope that this will be helpful to the author. In short it has to do with narrowing the scope, delimitting traces to be followed and make it more clear. Especially this concerns the two first sections and the last.

Introduction

The introduction needs to be revised in order to strengthen the readability and clarity.  It has to be shortened and edited according to; what is this article about, what kind of material is it based on and what do the reader need to know in order to be able to understand it. The information is there but it needs a structure. I suggest it is either divided into fractions or just shortened. It would also benefit from a short presentation of yoik and standing rock separately. (p1. check vuoiŋŋalašvuohta, it is spelled like I have spelled it here.)

Following sections:

1.       This section needs revision in order to make it clearer. I suggest that the author change the headline and divide the section in two or three sections. It has two main traces as I understand it. The first has to do with Sámi/indigenous colonial history and the second one is western versus indigenous research approaches. The reference to pedagogy of detachment is interesting, but it does not add anything to the paper, or one could argue, it adds one thing to many for the paper. I suggest that references like this are put aside in this paper, to be used in others.

It is enough to make a brief description of Sámi colonial history, with some main points. Even though it is a part of the colonial history of the world it is enough with a reference, since it is to little space to cover it all. I suggest two sections. One section that higlights the theoretical approach referring to e.g. Kuokkanen, and one with a brief and short overview of Sámi colonial history that is relevant for the paper.

In page 5: “ the substantial colonization of the Sámi population was not the geographical borders.” I am not sure whether this is the author’s opinion or Nergårds. The national borders has caused numerous conflicts within migrating reindeer herder groups and split families. Closing of borders has caused historical events e.g. like the forced moving of many Sámi families during the 20th century on the Swedish side of Sápmi, or the closing of the borders in middle of 19th century causing the Guovdageaidnu uprising- these are events still causing court cases today.

2.       This section needs a minor revision in order to strengthen the clarity. It has two main themes, one is Sámi spirituality in general and challenges connected with verbalizing (vocalization or verbalizing?), and western categories, the second is more specific about yoik in this connection. With some few sentences and maybe some rearranging this could be clearer.

3.       I suggest rearranging of the text. Oskals article on náhppi is a good analytical tool for this section and I suggest it comes first in the section and with a broader explanation. Next, about yoik and how its function has changed in Sámi society over time and tensions connected to moving it to new arenas. Last in this section could be Marielles experiences in her local community. This makes it more readable and also highlights the parallel to the evolving of the náhppi- from utility item to art. (That was very good!)

4.       I am a bit unsure about this section about identity and if it maybe should be a part of a new paper? Reading part 1,2, 3 it points out to the 5th section, Situating our bodies. This section has though to be connected to the former. Maybe some of the new theory that is mentioned here should be mentioned already in the first section?

5.       See point 4.

6.       In the closure it would be preferable to get a summary about the most important findings and what you have discovered about Sámi spirituality and yoik, and their relations in this paper. I also suggest that you highlight the fact that you are writing from a Sámi point of view (both here and in the beginning) using Sámi informants and research. This is very important, and makes your contribution strong.

This was an interesting paper to read. It revolves around themes as identity, yoik and spirituality.  The main material is contained by an interview with Marielle Gaup Beaska,  and use of Sámi research. The strongest aspect of the paper is its originality and the fact that it makes use of Sámi research, which is important in indigenous methodology but seldom done in practice. This paper carries a lot of information. With some sorting and a better structure it will contribute to the discourses about Sámi spirituality and yoik today, a topic that is less discussed from a Sámi point of view.

General comments:

Make sure to follow up theories and statements throughout the text.

Make sure the sections connect with each other.

Narrow down to a few traces, save the rest for coming papers. I suggest that the authors own struggling with identity could be put in another paper, since this would be relatively easy done. Interesting, but enough for a new paper.

It has some structural challenges. Because of the importance of the paper I have chosen to comment it quite thoroughly, and I hope that this will be helpful to the author. In short it has to do with narrowing the scope, delimitting traces to be followed and make it more clear. Especially this concerns the two first sections and the last.

Introduction

The introduction needs to be revised in order to strengthen the readability and clarity.  It has to be shortened and edited according to; what is this article about, what kind of material is it based on and what do the reader need to know in order to be able to understand it. The information is there but it needs a structure. I suggest it is either divided into fractions or just shortened. It would also benefit from a short presentation of yoik and standing rock separately. (p1. check vuoiŋŋalašvuohta, it is spelled like I have spelled it here.)

Following sections:

1.       This section needs revision in order to make it clearer. I suggest that the author change the headline and divide the section in two or three sections. It has two main traces as I understand it. The first has to do with Sámi/indigenous colonial history and the second one is western versus indigenous research approaches. The reference to pedagogy of detachment is interesting, but it does not add anything to the paper, or one could argue, it adds one thing to many for the paper. I suggest that references like this are put aside in this paper, to be used in others.

It is enough to make a brief description of Sámi colonial history, with some main points. Even though it is a part of the colonial history of the world it is enough with a reference, since it is to little space to cover it all. I suggest two sections. One section that higlights the theoretical approach referring to e.g. Kuokkanen, and one with a brief and short overview of Sámi colonial history that is relevant for the paper.

In page 5: “ the substantial colonization of the Sámi population was not the geographical borders.” I am not sure whether this is the author’s opinion or Nergårds. The national borders has caused numerous conflicts within migrating reindeer herder groups and split families. Closing of borders has caused historical events e.g. like the forced moving of many Sámi families during the 20th century on the Swedish side of Sápmi, or the closing of the borders in middle of 19th century causing the Guovdageaidnu uprising- these are events still causing court cases today.

2.       This section needs a minor revision in order to strengthen the clarity. It has two main themes, one is Sámi spirituality in general and challenges connected with verbalizing (vocalization or verbalizing?), and western categories, the second is more specific about yoik in this connection. With some few sentences and maybe some rearranging this could be clearer.

3.       I suggest rearranging of the text. Oskals article on náhppi is a good analytical tool for this section and I suggest it comes first in the section and with a broader explanation. Next, about yoik and how its function has changed in Sámi society over time and tensions connected to moving it to new arenas. Last in this section could be Marielles experiences in her local community. This makes it more readable and also highlights the parallel to the evolving of the náhppi- from utility item to art. (That was very good!)

4.       I am a bit unsure about this section about identity and if it maybe should be a part of a new paper? Reading part 1,2, 3 it points out to the 5th section, Situating our bodies. This section has though to be connected to the former. Maybe some of the new theory that is mentioned here should be mentioned already in the first section?

5.       See point 4.

6.       In the closure it would be preferable to get a summary about the most important findings and what you have discovered about Sámi spirituality and yoik, and their relations in this paper. I also suggest that you highlight the fact that you are writing from a Sámi point of view (both here and in the beginning) using Sámi informants and research. This is very important, and makes your contribution strong.

This was an interesting paper to read. It revolves around themes as identity, yoik and spirituality.  The main material is contained by an interview with Marielle Gaup Beaska,  and use of Sámi research. The strongest aspect of the paper is its originality and the fact that it makes use of Sámi research, which is important in indigenous methodology but seldom done in practice. This paper carries a lot of information. With some sorting and a better structure it will contribute to the discourses about Sámi spirituality and yoik today, a topic that is less discussed from a Sámi point of view.

General comments:

Make sure to follow up theories and statements throughout the text.

Make sure the sections connect with each other.

Narrow down to a few traces, save the rest for coming papers. I suggest that the authors own struggling with identity could be put in another paper, since this would be relatively easy done. Interesting, but enough for a new paper.

It has some structural challenges. Because of the importance of the paper I have chosen to comment it quite thoroughly, and I hope that this will be helpful to the author. In short it has to do with narrowing the scope, delimitting traces to be followed and make it more clear. Especially this concerns the two first sections and the last.

Introduction

The introduction needs to be revised in order to strengthen the readability and clarity.  It has to be shortened and edited according to; what is this article about, what kind of material is it based on and what do the reader need to know in order to be able to understand it. The information is there but it needs a structure. I suggest it is either divided into fractions or just shortened. It would also benefit from a short presentation of yoik and standing rock separately. (p1. check vuoiŋŋalašvuohta, it is spelled like I have spelled it here.)

Following sections:

1.       This section needs revision in order to make it clearer. I suggest that the author change the headline and divide the section in two or three sections. It has two main traces as I understand it. The first has to do with Sámi/indigenous colonial history and the second one is western versus indigenous research approaches. The reference to pedagogy of detachment is interesting, but it does not add anything to the paper, or one could argue, it adds one thing to many for the paper. I suggest that references like this are put aside in this paper, to be used in others.

It is enough to make a brief description of Sámi colonial history, with some main points. Even though it is a part of the colonial history of the world it is enough with a reference, since it is to little space to cover it all. I suggest two sections. One section that higlights the theoretical approach referring to e.g. Kuokkanen, and one with a brief and short overview of Sámi colonial history that is relevant for the paper.

In page 5: “ the substantial colonization of the Sámi population was not the geographical borders.” I am not sure whether this is the author’s opinion or Nergårds. The national borders has caused numerous conflicts within migrating reindeer herder groups and split families. Closing of borders has caused historical events e.g. like the forced moving of many Sámi families during the 20th century on the Swedish side of Sápmi, or the closing of the borders in middle of 19th century causing the Guovdageaidnu uprising- these are events still causing court cases today.

2.       This section needs a minor revision in order to strengthen the clarity. It has two main themes, one is Sámi spirituality in general and challenges connected with verbalizing (vocalization or verbalizing?), and western categories, the second is more specific about yoik in this connection. With some few sentences and maybe some rearranging this could be clearer.

3.       I suggest rearranging of the text. Oskals article on náhppi is a good analytical tool for this section and I suggest it comes first in the section and with a broader explanation. Next, about yoik and how its function has changed in Sámi society over time and tensions connected to moving it to new arenas. Last in this section could be Marielles experiences in her local community. This makes it more readable and also highlights the parallel to the evolving of the náhppi- from utility item to art. (That was very good!)

4.       I am a bit unsure about this section about identity and if it maybe should be a part of a new paper? Reading part 1,2, 3 it points out to the 5th section, Situating our bodies. This section has though to be connected to the former. Maybe some of the new theory that is mentioned here should be mentioned already in the first section?

5.       See point 4.

6.       In the closure it would be preferable to get a summary about the most important findings and what you have discovered about Sámi spirituality and yoik, and their relations in this paper. I also suggest that you highlight the fact that you are writing from a Sámi point of view (both here and in the beginning) using Sámi informants and research. This is very important, and makes your contribution strong.

This was an interesting paper to read. It revolves around themes as identity, yoik and spirituality.  The main material is contained by an interview with Marielle Gaup Beaska,  and use of Sámi research. The strongest aspect of the paper is its originality and the fact that it makes use of Sámi research, which is important in indigenous methodology but seldom done in practice. This paper carries a lot of information. With some sorting and a better structure it will contribute to the discourses about Sámi spirituality and yoik today, a topic that is less discussed from a Sámi point of view.

General comments:

Make sure to follow up theories and statements throughout the text.

Make sure the sections connect with each other.

Narrow down to a few traces, save the rest for coming papers. I suggest that the authors own struggling with identity could be put in another paper, since this would be relatively easy done. Interesting, but enough for a new paper.

It has some structural challenges. Because of the importance of the paper I have chosen to comment it quite thoroughly, and I hope that this will be helpful to the author. In short it has to do with narrowing the scope, delimitting traces to be followed and make it more clear. Especially this concerns the two first sections and the last.

Introduction

The introduction needs to be revised in order to strengthen the readability and clarity.  It has to be shortened and edited according to; what is this article about, what kind of material is it based on and what do the reader need to know in order to be able to understand it. The information is there but it needs a structure. I suggest it is either divided into fractions or just shortened. It would also benefit from a short presentation of yoik and standing rock separately. (p1. check vuoiŋŋalašvuohta, it is spelled like I have spelled it here.)

Following sections:

1.       This section needs revision in order to make it clearer. I suggest that the author change the headline and divide the section in two or three sections. It has two main traces as I understand it. The first has to do with Sámi/indigenous colonial history and the second one is western versus indigenous research approaches. The reference to pedagogy of detachment is interesting, but it does not add anything to the paper, or one could argue, it adds one thing to many for the paper. I suggest that references like this are put aside in this paper, to be used in others.

It is enough to make a brief description of Sámi colonial history, with some main points. Even though it is a part of the colonial history of the world it is enough with a reference, since it is to little space to cover it all. I suggest two sections. One section that higlights the theoretical approach referring to e.g. Kuokkanen, and one with a brief and short overview of Sámi colonial history that is relevant for the paper.

In page 5: “ the substantial colonization of the Sámi population was not the geographical borders.” I am not sure whether this is the author’s opinion or Nergårds. The national borders has caused numerous conflicts within migrating reindeer herder groups and split families. Closing of borders has caused historical events e.g. like the forced moving of many Sámi families during the 20th century on the Swedish side of Sápmi, or the closing of the borders in middle of 19th century causing the Guovdageaidnu uprising- these are events still causing court cases today.

2.       This section needs a minor revision in order to strengthen the clarity. It has two main themes, one is Sámi spirituality in general and challenges connected with verbalizing (vocalization or verbalizing?), and western categories, the second is more specific about yoik in this connection. With some few sentences and maybe some rearranging this could be clearer.

3.       I suggest rearranging of the text. Oskals article on náhppi is a good analytical tool for this section and I suggest it comes first in the section and with a broader explanation. Next, about yoik and how its function has changed in Sámi society over time and tensions connected to moving it to new arenas. Last in this section could be Marielles experiences in her local community. This makes it more readable and also highlights the parallel to the evolving of the náhppi- from utility item to art. (That was very good!)

4.       I am a bit unsure about this section about identity and if it maybe should be a part of a new paper? Reading part 1,2, 3 it points out to the 5th section, Situating our bodies. This section has though to be connected to the former. Maybe some of the new theory that is mentioned here should be mentioned already in the first section?

5.       See point 4.

6.       In the closure it would be preferable to get a summary about the most important findings and what you have discovered about Sámi spirituality and yoik, and their relations in this paper. I also suggest that you highlight the fact that you are writing from a Sámi point of view (both here and in the beginning) using Sámi informants and research. This is very important, and makes your contribution strong.

This was an interesting paper to read. It revolves around themes as identity, yoik and spirituality.  The main material is contained by an interview with Marielle Gaup Beaska,  and use of Sámi research. The strongest aspect of the paper is its originality and the fact that it makes use of Sámi research, which is important in indigenous methodology but seldom done in practice. This paper carries a lot of information. With some sorting and a better structure it will contribute to the discourses about Sámi spirituality and yoik today, a topic that is less discussed from a Sámi point of view.

General comments:

Make sure to follow up theories and statements throughout the text.

Make sure the sections connect with each other.

Narrow down to a few traces, save the rest for coming papers. I suggest that the authors own struggling with identity could be put in another paper, since this would be relatively easy done. Interesting, but enough for a new paper.

It has some structural challenges. Because of the importance of the paper I have chosen to comment it quite thoroughly, and I hope that this will be helpful to the author. In short it has to do with narrowing the scope, delimitting traces to be followed and make it more clear. Especially this concerns the two first sections and the last.

Introduction

The introduction needs to be revised in order to strengthen the readability and clarity.  It has to be shortened and edited according to; what is this article about, what kind of material is it based on and what do the reader need to know in order to be able to understand it. The information is there but it needs a structure. I suggest it is either divided into fractions or just shortened. It would also benefit from a short presentation of yoik and standing rock separately. (p1. check vuoiŋŋalašvuohta, it is spelled like I have spelled it here.)

Following sections:

1.       This section needs revision in order to make it clearer. I suggest that the author change the headline and divide the section in two or three sections. It has two main traces as I understand it. The first has to do with Sámi/indigenous colonial history and the second one is western versus indigenous research approaches. The reference to pedagogy of detachment is interesting, but it does not add anything to the paper, or one could argue, it adds one thing to many for the paper. I suggest that references like this are put aside in this paper, to be used in others.

It is enough to make a brief description of Sámi colonial history, with some main points. Even though it is a part of the colonial history of the world it is enough with a reference, since it is to little space to cover it all. I suggest two sections. One section that higlights the theoretical approach referring to e.g. Kuokkanen, and one with a brief and short overview of Sámi colonial history that is relevant for the paper.

In page 5: “ the substantial colonization of the Sámi population was not the geographical borders.” I am not sure whether this is the author’s opinion or Nergårds. The national borders has caused numerous conflicts within migrating reindeer herder groups and split families. Closing of borders has caused historical events e.g. like the forced moving of many Sámi families during the 20th century on the Swedish side of Sápmi, or the closing of the borders in middle of 19th century causing the Guovdageaidnu uprising- these are events still causing court cases today.

2.       This section needs a minor revision in order to strengthen the clarity. It has two main themes, one is Sámi spirituality in general and challenges connected with verbalizing (vocalization or verbalizing?), and western categories, the second is more specific about yoik in this connection. With some few sentences and maybe some rearranging this could be clearer.

3.       I suggest rearranging of the text. Oskals article on náhppi is a good analytical tool for this section and I suggest it comes first in the section and with a broader explanation. Next, about yoik and how its function has changed in Sámi society over time and tensions connected to moving it to new arenas. Last in this section could be Marielles experiences in her local community. This makes it more readable and also highlights the parallel to the evolving of the náhppi- from utility item to art. (That was very good!)

4.       I am a bit unsure about this section about identity and if it maybe should be a part of a new paper? Reading part 1,2, 3 it points out to the 5th section, Situating our bodies. This section has though to be connected to the former. Maybe some of the new theory that is mentioned here should be mentioned already in the first section?

5.       See point 4.

6.       In the closure it would be preferable to get a summary about the most important findings and what you have discovered about Sámi spirituality and yoik, and their relations in this paper. I also suggest that you highlight the fact that you are writing from a Sámi point of view (both here and in the beginning) using Sámi informants and research. This is very important, and makes your contribution strong.

This was an interesting paper to read. It revolves around themes as identity, yoik and spirituality.  The main material is contained by an interview with Marielle Gaup Beaska,  and use of Sámi research. The strongest aspect of the paper is its originality and the fact that it makes use of Sámi research, which is important in indigenous methodology but seldom done in practice. This paper carries a lot of information. With some sorting and a better structure it will contribute to the discourses about Sámi spirituality and yoik today, a topic that is less discussed from a Sámi point of view.

General comments:

Make sure to follow up theories and statements throughout the text.

Make sure the sections connect with each other.

Narrow down to a few traces, save the rest for coming papers. I suggest that the authors own struggling with identity could be put in another paper, since this would be relatively easy done. Interesting, but enough for a new paper.

It has some structural challenges. Because of the importance of the paper I have chosen to comment it quite thoroughly, and I hope that this will be helpful to the author. In short it has to do with narrowing the scope, delimitting traces to be followed and make it more clear. Especially this concerns the two first sections and the last.

Introduction

The introduction needs to be revised in order to strengthen the readability and clarity.  It has to be shortened and edited according to; what is this article about, what kind of material is it based on and what do the reader need to know in order to be able to understand it. The information is there but it needs a structure. I suggest it is either divided into fractions or just shortened. It would also benefit from a short presentation of yoik and standing rock separately. (p1. check vuoiŋŋalašvuohta, it is spelled like I have spelled it here.)

Following sections:

1.       This section needs revision in order to make it clearer. I suggest that the author change the headline and divide the section in two or three sections. It has two main traces as I understand it. The first has to do with Sámi/indigenous colonial history and the second one is western versus indigenous research approaches. The reference to pedagogy of detachment is interesting, but it does not add anything to the paper, or one could argue, it adds one thing to many for the paper. I suggest that references like this are put aside in this paper, to be used in others.

It is enough to make a brief description of Sámi colonial history, with some main points. Even though it is a part of the colonial history of the world it is enough with a reference, since it is to little space to cover it all. I suggest two sections. One section that higlights the theoretical approach referring to e.g. Kuokkanen, and one with a brief and short overview of Sámi colonial history that is relevant for the paper.

In page 5: “ the substantial colonization of the Sámi population was not the geographical borders.” I am not sure whether this is the author’s opinion or Nergårds. The national borders has caused numerous conflicts within migrating reindeer herder groups and split families. Closing of borders has caused historical events e.g. like the forced moving of many Sámi families during the 20th century on the Swedish side of Sápmi, or the closing of the borders in middle of 19th century causing the Guovdageaidnu uprising- these are events still causing court cases today.

2.       This section needs a minor revision in order to strengthen the clarity. It has two main themes, one is Sámi spirituality in general and challenges connected with verbalizing (vocalization or verbalizing?), and western categories, the second is more specific about yoik in this connection. With some few sentences and maybe some rearranging this could be clearer.

3.       I suggest rearranging of the text. Oskals article on náhppi is a good analytical tool for this section and I suggest it comes first in the section and with a broader explanation. Next, about yoik and how its function has changed in Sámi society over time and tensions connected to moving it to new arenas. Last in this section could be Marielles experiences in her local community. This makes it more readable and also highlights the parallel to the evolving of the náhppi- from utility item to art. (That was very good!)

4.       I am a bit unsure about this section about identity and if it maybe should be a part of a new paper? Reading part 1,2, 3 it points out to the 5th section, Situating our bodies. This section has though to be connected to the former. Maybe some of the new theory that is mentioned here should be mentioned already in the first section?

5.       See point 4.

6.       In the closure it would be preferable to get a summary about the most important findings and what you have discovered about Sámi spirituality and yoik, and their relations in this paper. I also suggest that you highlight the fact that you are writing from a Sámi point of view (both here and in the beginning) using Sámi informants and research. This is very important, and makes your contribution strong.

This was an interesting paper to read. It revolves around themes as identity, yoik and spirituality.  The main material is contained by an interview with Marielle Gaup Beaska,  and use of Sámi research. The strongest aspect of the paper is its originality and the fact that it makes use of Sámi research, which is important in indigenous methodology but seldom done in practice. This paper carries a lot of information. With some sorting and a better structure it will contribute to the discourses about Sámi spirituality and yoik today, a topic that is less discussed from a Sámi point of view.

General comments:

Make sure to follow up theories and statements throughout the text.

Make sure the sections connect with each other.

Narrow down to a few traces, save the rest for coming papers. I suggest that the authors own struggling with identity could be put in another paper, since this would be relatively easy done. Interesting, but enough for a new paper.

It has some structural challenges. Because of the importance of the paper I have chosen to comment it quite thoroughly, and I hope that this will be helpful to the author. In short it has to do with narrowing the scope, delimitting traces to be followed and make it more clear. Especially this concerns the two first sections and the last.

Introduction

The introduction needs to be revised in order to strengthen the readability and clarity.  It has to be shortened and edited according to; what is this article about, what kind of material is it based on and what do the reader need to know in order to be able to understand it. The information is there but it needs a structure. I suggest it is either divided into fractions or just shortened. It would also benefit from a short presentation of yoik and standing rock separately. (p1. check vuoiŋŋalašvuohta, it is spelled like I have spelled it here.)

Following sections:

1.       This section needs revision in order to make it clearer. I suggest that the author change the headline and divide the section in two or three sections. It has two main traces as I understand it. The first has to do with Sámi/indigenous colonial history and the second one is western versus indigenous research approaches. The reference to pedagogy of detachment is interesting, but it does not add anything to the paper, or one could argue, it adds one thing to many for the paper. I suggest that references like this are put aside in this paper, to be used in others.

It is enough to make a brief description of Sámi colonial history, with some main points. Even though it is a part of the colonial history of the world it is enough with a reference, since it is to little space to cover it all. I suggest two sections. One section that higlights the theoretical approach referring to e.g. Kuokkanen, and one with a brief and short overview of Sámi colonial history that is relevant for the paper.

In page 5: “ the substantial colonization of the Sámi population was not the geographical borders.” I am not sure whether this is the author’s opinion or Nergårds. The national borders has caused numerous conflicts within migrating reindeer herder groups and split families. Closing of borders has caused historical events e.g. like the forced moving of many Sámi families during the 20th century on the Swedish side of Sápmi, or the closing of the borders in middle of 19th century causing the Guovdageaidnu uprising- these are events still causing court cases today.

2.       This section needs a minor revision in order to strengthen the clarity. It has two main themes, one is Sámi spirituality in general and challenges connected with verbalizing (vocalization or verbalizing?), and western categories, the second is more specific about yoik in this connection. With some few sentences and maybe some rearranging this could be clearer.

3.       I suggest rearranging of the text. Oskals article on náhppi is a good analytical tool for this section and I suggest it comes first in the section and with a broader explanation. Next, about yoik and how its function has changed in Sámi society over time and tensions connected to moving it to new arenas. Last in this section could be Marielles experiences in her local community. This makes it more readable and also highlights the parallel to the evolving of the náhppi- from utility item to art. (That was very good!)

4.       I am a bit unsure about this section about identity and if it maybe should be a part of a new paper? Reading part 1,2, 3 it points out to the 5th section, Situating our bodies. This section has though to be connected to the former. Maybe some of the new theory that is mentioned here should be mentioned already in the first section?

5.       See point 4.

6.       In the closure it would be preferable to get a summary about the most important findings and what you have discovered about Sámi spirituality and yoik, and their relations in this paper. I also suggest that you highlight the fact that you are writing from a Sámi point of view (both here and in the beginning) using Sámi informants and research. This is very important, and makes your contribution strong.

This was an interesting paper to read. It revolves around themes as identity, yoik and spirituality.  The main material is contained by an interview with Marielle Gaup Beaska,  and use of Sámi research. The strongest aspect of the paper is its originality and the fact that it makes use of Sámi research, which is important in indigenous methodology but seldom done in practice. This paper carries a lot of information. With some sorting and a better structure it will contribute to the discourses about Sámi spirituality and yoik today, a topic that is less discussed from a Sámi point of view.

General comments:

Make sure to follow up theories and statements throughout the text.

Make sure the sections connect with each other.

Narrow down to a few traces, save the rest for coming papers. I suggest that the authors own struggling with identity could be put in another paper, since this would be relatively easy done. Interesting, but enough for a new paper.

It has some structural challenges. Because of the importance of the paper I have chosen to comment it quite thoroughly, and I hope that this will be helpful to the author. In short it has to do with narrowing the scope, delimitting traces to be followed and make it more clear. Especially this concerns the two first sections and the last.

Introduction

The introduction needs to be revised in order to strengthen the readability and clarity.  It has to be shortened and edited according to; what is this article about, what kind of material is it based on and what do the reader need to know in order to be able to understand it. The information is there but it needs a structure. I suggest it is either divided into fractions or just shortened. It would also benefit from a short presentation of yoik and standing rock separately. (p1. check vuoiŋŋalašvuohta, it is spelled like I have spelled it here.)

Following sections:

1.       This section needs revision in order to make it clearer. I suggest that the author change the headline and divide the section in two or three sections. It has two main traces as I understand it. The first has to do with Sámi/indigenous colonial history and the second one is western versus indigenous research approaches. The reference to pedagogy of detachment is interesting, but it does not add anything to the paper, or one could argue, it adds one thing to many for the paper. I suggest that references like this are put aside in this paper, to be used in others.

It is enough to make a brief description of Sámi colonial history, with some main points. Even though it is a part of the colonial history of the world it is enough with a reference, since it is to little space to cover it all. I suggest two sections. One section that higlights the theoretical approach referring to e.g. Kuokkanen, and one with a brief and short overview of Sámi colonial history that is relevant for the paper.

In page 5: “ the substantial colonization of the Sámi population was not the geographical borders.” I am not sure whether this is the author’s opinion or Nergårds. The national borders has caused numerous conflicts within migrating reindeer herder groups and split families. Closing of borders has caused historical events e.g. like the forced moving of many Sámi families during the 20th century on the Swedish side of Sápmi, or the closing of the borders in middle of 19th century causing the Guovdageaidnu uprising- these are events still causing court cases today.

2.       This section needs a minor revision in order to strengthen the clarity. It has two main themes, one is Sámi spirituality in general and challenges connected with verbalizing (vocalization or verbalizing?), and western categories, the second is more specific about yoik in this connection. With some few sentences and maybe some rearranging this could be clearer.

3.       I suggest rearranging of the text. Oskals article on náhppi is a good analytical tool for this section and I suggest it comes first in the section and with a broader explanation. Next, about yoik and how its function has changed in Sámi society over time and tensions connected to moving it to new arenas. Last in this section could be Marielles experiences in her local community. This makes it more readable and also highlights the parallel to the evolving of the náhppi- from utility item to art. (That was very good!)

4.       I am a bit unsure about this section about identity and if it maybe should be a part of a new paper? Reading part 1,2, 3 it points out to the 5th section, Situating our bodies. This section has though to be connected to the former. Maybe some of the new theory that is mentioned here should be mentioned already in the first section?

5.       See point 4.

6.       In the closure it would be preferable to get a summary about the most important findings and what you have discovered about Sámi spirituality and yoik, and their relations in this paper. I also suggest that you highlight the fact that you are writing from a Sámi point of view (both here and in the beginning) using Sámi informants and research. This is very important, and makes your contribution strong.

This was an interesting paper to read. It revolves around themes as identity, yoik and spirituality.  The main material is contained by an interview with Marielle Gaup Beaska,  and use of Sámi research. The strongest aspect of the paper is its originality and the fact that it makes use of Sámi research, which is important in indigenous methodology but seldom done in practice. This paper carries a lot of information. With some sorting and a better structure it will contribute to the discourses about Sámi spirituality and yoik today, a topic that is less discussed from a Sámi point of view.

General comments:

Make sure to follow up theories and statements throughout the text.

Make sure the sections connect with each other.

Narrow down to a few traces, save the rest for coming papers. I suggest that the authors own struggling with identity could be put in another paper, since this would be relatively easy done. Interesting, but enough for a new paper.

It has some structural challenges. Because of the importance of the paper I have chosen to comment it quite thoroughly, and I hope that this will be helpful to the author. In short it has to do with narrowing the scope, delimitting traces to be followed and make it more clear. Especially this concerns the two first sections and the last.

Introduction

The introduction needs to be revised in order to strengthen the readability and clarity.  It has to be shortened and edited according to; what is this article about, what kind of material is it based on and what do the reader need to know in order to be able to understand it. The information is there but it needs a structure. I suggest it is either divided into fractions or just shortened. It would also benefit from a short presentation of yoik and standing rock separately. (p1. check vuoiŋŋalašvuohta, it is spelled like I have spelled it here.)

Following sections:

1.       This section needs revision in order to make it clearer. I suggest that the author change the headline and divide the section in two or three sections. It has two main traces as I understand it. The first has to do with Sámi/indigenous colonial history and the second one is western versus indigenous research approaches. The reference to pedagogy of detachment is interesting, but it does not add anything to the paper, or one could argue, it adds one thing to many for the paper. I suggest that references like this are put aside in this paper, to be used in others.

It is enough to make a brief description of Sámi colonial history, with some main points. Even though it is a part of the colonial history of the world it is enough with a reference, since it is to little space to cover it all. I suggest two sections. One section that higlights the theoretical approach referring to e.g. Kuokkanen, and one with a brief and short overview of Sámi colonial history that is relevant for the paper.

In page 5: “ the substantial colonization of the Sámi population was not the geographical borders.” I am not sure whether this is the author’s opinion or Nergårds. The national borders has caused numerous conflicts within migrating reindeer herder groups and split families. Closing of borders has caused historical events e.g. like the forced moving of many Sámi families during the 20th century on the Swedish side of Sápmi, or the closing of the borders in middle of 19th century causing the Guovdageaidnu uprising- these are events still causing court cases today.

2.       This section needs a minor revision in order to strengthen the clarity. It has two main themes, one is Sámi spirituality in general and challenges connected with verbalizing (vocalization or verbalizing?), and western categories, the second is more specific about yoik in this connection. With some few sentences and maybe some rearranging this could be clearer.

3.       I suggest rearranging of the text. Oskals article on náhppi is a good analytical tool for this section and I suggest it comes first in the section and with a broader explanation. Next, about yoik and how its function has changed in Sámi society over time and tensions connected to moving it to new arenas. Last in this section could be Marielles experiences in her local community. This makes it more readable and also highlights the parallel to the evolving of the náhppi- from utility item to art. (That was very good!)

4.       I am a bit unsure about this section about identity and if it maybe should be a part of a new paper? Reading part 1,2, 3 it points out to the 5th section, Situating our bodies. This section has though to be connected to the former. Maybe some of the new theory that is mentioned here should be mentioned already in the first section?

5.       See point 4.

6.       In the closure it would be preferable to get a summary about the most important findings and what you have discovered about Sámi spirituality and yoik, and their relations in this paper. I also suggest that you highlight the fact that you are writing from a Sámi point of view (both here and in the beginning) using Sámi informants and research. This is very important, and makes your contribution strong.

Author Response

General comments:

Make sure to follow up theories and statements throughout the text.

Done

Make sure the sections connect with each other.

Done

Narrow down to a few traces, save the rest for coming papers. I suggest that the authors own struggling with identity could be put in another paper, since this would be relatively easy done. Interesting, but enough for a new paper.

I have narrowed down, and minimised some of the section about identity. However as the theme is identity and spirituality I have kept some parts of it. But let me know if this is not working.

It has some structural challenges. Because of the importance of the paper I have chosen to comment it quite thoroughly, and I hope that this will be helpful to the author. In short it has to do with narrowing the scope, delimitting traces to be followed and make it more clear. Especially this concerns the two first sections and the last.

Thanks for being so thorough, I appreciate it.

Introduction

The introduction needs to be revised in order to strengthen the readability and clarity. It has to be shortened and edited according to; what is this article about, what kind of material is it based on and what do the reader need to know in order to be able to understand it. The information is there but it needs a structure. I suggest it is either divided into fractions or just shortened. It would also benefit from a short presentation of yoik and standing rock separately. (p1. check vuoiŋŋalašvuohta, it is spelled like I have spelled it here.)

I have structured it and divided it in sections and included a section about Standing Rock and joik. I have also added two research questions.

Following sections:

1. This section needs revision in order to make it clearer. I suggest that the author change the headline and divide the section in two or three sections. It has two main traces as I understand it. The first has to do with Sámi/indigenous colonial history and the second one is western versus indigenous research approaches. The reference to pedagogy of detachment is interesting, but it does not add anything to the paper, or one could argue, it adds one thing to many for the paper. I suggest that references like this are put aside in this paper, to be used in others.

I have revised it and tried to incorporate the section about the paradigm of detachment because it is so central to the theme of the paper. Let me know it it still seems superfluous.

It is enough to make a brief description of Sámi colonial history, with some main points. Even though it is a part of the colonial history of the world it is enough with a reference, since it is to little space to cover it all. I suggest two sections. One section that higlights the theoretical approach referring to e.g. Kuokkanen, and one with a brief and short overview of Sámi colonial history that is relevant for the paper.

I have restructured this part too.

In page 5: “ the substantial colonization of the Sámi population was not the geographical borders.” I am not sure whether this is the author’s opinion or Nergårds. The national borders has caused numerous conflicts within migrating reindeer herder groups and split families. Closing of borders has caused historical events e.g. like the forced moving of many Sámi families during the 20th century on the Swedish side of Sápmi, or the closing of the borders in middle of 19th century causing the Guovdageaidnu uprising- these are events still causing court cases today.

This is true, i have made sure it is clear that it is Nergårds opinion.

2. This section needs a minor revision in order to strengthen the clarity. It has two main themes, one is Sámi spirituality in general and challenges connected with verbalizing (vocalization or verbalizing?), and western categories, the second is more specific about yoik in this connection. With some few sentences and maybe some rearranging this could be clearer.

I hope this section seems clearer now.

3. I suggest rearranging of the text. Oskals article on náhppi is a good analytical tool for this section and I suggest it comes first in the section and with a broader explanation. Next, about yoik and how its function has changed in Sámi society over time and tensions connected to moving it to new arenas. Last in this section could be Marielles experiences in her local community. This makes it more readable and also highlights the parallel to the evolving of the náhppi- from utility item to art. (That was very good!)

I have rearranged the text as you advised, however it was not possible to restructure it completely.

4. I am a bit unsure about this section about identity and if it maybe should be a part of a new paper? Reading part 1,2, 3 it points out to the 5th section, Situating our bodies. This section has though to be connected to the former. Maybe some of the new theory that is mentioned here should be mentioned already in the first section?

I have moved it upwards and shortened it.

5. See point 4.

6. In the closure it would be preferable to get a summary about the most important findings and what you have discovered about Sámi spirituality and yoik, and their relations in this paper. I also suggest that you highlight the fact that you are writing from a Sámi point of view (both here and in the beginning) using Sámi informants and research. This is very important, and makes your contribution strong.

Yes, thank you!

Round 2

Reviewer 2 Report

I do not have any further suggestions and am fond of improvements that have been made carefully.